# 'Cam Girls and Adult Performers Are Enjoying a Boom in Business': The Reportage on the Pandemic Impact on Virtual Sex Work

Valeria Rubattu *, Alicja Perdion and Belinda Brooks-Gordon

Department of Psychological Sciences, Birkbeck University of London, Malet Street, Bloomsbury, London WC1E 7HX, UK
* Correspondence: valeriarubattu91@outlook.com

**Abstract: Introduction:** Webcamming as a digital practice has increased in popularity over the last decade. With the outbreak COVID-19 and lockdowns across the globe, cam sites experienced an upsurge in both performers and viewers, and the main platform *OnlyFans*, increased its market share and saturation. The objective of this study was to explore the perceived impact of the COVID-19 pandemic, and subsequent economic hardship, on indirect and digitally mediated sex work. In doing so, it also explored the mediatisation of the creators of erotic content and their marketing on *OnlyFans*. **Method:** Data was collected from news media outlets on the effects of the outbreak of the virus on the online sex industry. Mainstream media news articles (N = 40) were drawn from 19 different sources that discussed changes occurring in the digitally mediated sex market during the COVID-19 pandemic. The data was drawn from across the political spectrum and type of media source to include broadsheet, tabloid, and regional news as well as broadcast media. The dataset was divided into two and independently analysed by two different researchers analysing 20 sources each. Analysis was conducted using Grounded Theory, an inductive approach frequently used due to aid concept development, as the aim was to develop theory on the mediatisation of the experiences and process of virtual sex work without drawing on sex workers' own resources at a time crucial to their income. **Results:** The findings revealed reportage of increased engagement in digital sex work in three areas: expansion of the online sex service sites; new digital sex workers joining the industry; and those who provided online sex services prior to the pandemic. A continuum of experiences emerged and the results show how online sex workers reportedly monetised the loneliness of clients and how new fetishes such as illness and Covid fetishes emerged. **Conclusions:** Given the remarkable success of adult websites amid the pandemic, this research provides new evidence on the reportage of the use of cam sites, and OnlyFans in particular. The findings provides new data on how digital sex workers' experiences were represented during the pandemic and reveal a nuanced picture behind the upsurge in online work. News media outlets are crucial in the social construction of online sex work and have the power to affect peoples' perception of this work. Additionally, press articles can provide a space where sex workers' voices can be heard. It is therefore a key area to examine in relation to the public opinion of sex work, which in turn affects public policy, and its decriminalisation and eventual destigmatisation. These findings add to our understanding of erotic services and contribute to the growing literature on the mediatization of sex work. The study contributes new knowledge to the topic although further investigation is needed to achieve potential mainstreaming and destigmatisation for digital sex workers.

**Keywords:** sex work; digital; virtual; webcam; COVID; pandemic



## 1. Introduction

The COVID-19 pandemic and subsequent lockdowns introduced travel restrictions and the closure of non-essential workplaces, forcing people to stay at home to prevent the

spread of the virus (Döring 2020). While lockdowns were detrimental for many businesses in the service sector, the digital industries, including digital sex industries, experienced substantial growth (Martinez Dy and Jayawarna 2020). Webcamming was one of the areas through which traditional sex work had already expanded digitally (Henry and Farvid 2017) and it has been estimated to be a multibillion-dollar industry with individuals involved in this form of sex work referred to as 'webcam models' (Jones 2020). Their services include provision of online 'shows', typically of a sexual nature, which often incorporate elements of intimate conversation with their clientele as well as erotic striptease and explicit sex acts (Sanders et al. 2020; Jones 2020). These shows are live streamed via a webcam to customers' personal devices (Sanders et al. 2020).

Webcam models' performances differ from traditional pornography in that they are interactive; customers communicate with the performer throughout the show which allows the worker to tailor their performance according to viewers' desires (Jones 2020). Webcam platforms provide an interface between the worker and their viewers and operate with different business models. Some platforms require customers to pay per minute in order to watch the show, others are free but rely on customer's tipping the model throughout their show using tokens purchased from the platform (Sanders et al. 2018). The cam sites then profit by taking a percentage of the model's earnings (Matolcsi et al. 2021). When the webcam site has public access, the performers tend to post a topic (e.g., 1000 tokens for a specific sex act) by initiating a 'countdown' during which the viewers collectively tip the required number of tokens for the associated show (Jones 2020). One platform that experienced a particular spike in traffic that impacted the work of its content creators was *OnlyFans*—a website that also offers adult content and has gained market dominance (Lykousas et al. 2020; Lines 2020). The pandemic represented a massive structural intervention that impacted on public health, individual economic circumstances, and elevated the risk of social disadvantage for millions of people. The current research explores how the pandemic was reported to affect the work and lives of online sex workers engaged in webcamming or on sites such as *OnlyFans* by exploring the media reportage of the phenomenon in an unobtrusive way.

## 1.1. Emotional Labour

Studies into webcamming reveal that this form of online sex work attracts more women than men (Sanders et al. 2018; Jones 2020) and that webcam models tend to be younger than independent escorts (Sanders et al. 2018). According to Jones (2020), motivations for becoming a webcam model often include desires for exploring one's own sexuality and experiencing pleasure, in addition to autonomy and decent income. Webcam shows often revolve around interactive conversation between the model and their viewers, and cam models tend to perform different practices at different levels of eroticism, sometimes by removing only some clothing, or dancing, reading, or chatting with viewers (Henry and Farvid 2017); emotional labour can therefore be as important as the sexual aspect of this job (Sanders et al. 2020; Jones 2020).

Emotional labour consists of the individuals' effort to engage in surface acting (i.e., displayed emotions such as smiling), or deep acting (which includes regulating internal emotions and controlling pleasure or disgust) (Hochschild 1983). Nayar (2017) suggests that such strategies to help manage emotions guarantee 'professional immunity' from comments, either compliments or insults, that can affect the individual in a real-life setting. Achieving success within this industry has been shown to largely depend on the model's good communicative skills and availability to keep the viewers entertained throughout the show as well as creativity and readiness to experiment, rather than simply their good looks (Sanders et al. 2020; Van Doorn and Velthuis 2018). By contrast, Jones (2016) argues that camming is a real sexual digital interaction that does not require 'acting' during the performance. The encounters between performers and viewer, therefore, do not require emotional labour. Such encounters, Jones (2020) argues, do not always consist of sexual acts and often involve conversations with clients and culminate in the emergence of friendships,

with performers and viewers talking after the camera is turned off. Such notions of authenticity were developed by Parvez (2006) who provided reconstructed theory of emotional labour from the perspective of the consumer which enables a deeper exploration of women's ambivalence and the subsequent implications for understanding sexuality. More recently, it has been argued by Flubacher (2019) that this is now so widespread in the commercial world that 'selling the self' in the labour market is now a common expectation across many labour markets.

Scholars have explored the possible reasons for the absence of emotional labour in webcamming research. According to Lee (2021), cam models are not forced to engage in emotional labour to satisfy the clients because webcamming is a safer and more empowering labour when compared to other sex practices. The divergence in the past findings of the role of emotional labour in webcamming could be associated with the individual experiences of the performers or their relationships with the viewers, and it is clear that reports of whether and when cam models use such strategy are ripe for investigation.

### 1.2. Deconstructing 'Empowerment'

The relationship between emotional labour and empowerment was made by Hall (1995) who argued that to be in a position of power, women involved in telephone fantasy lines were able to create a language based on stereotypes of them being powerless. The reason the workers were in the erotic calls industry was due to not having access to other employment. To people outside the industry, workers within it are seen as powerless, yet they felt empowered both financially and by the freedom it offered. In this way, women were able to have one of the privileges of patriarchy, that of economic power. This feeling of empowerment is not unaccompanied by challenges. For example, Henry (2018) investigated the different perspectives of digital sex workers on a group of cam models in New Zealand, finding that the participants generally considered camming an unsafe practice after experiencing negative events, such as stalking or harassment (Henry 2018). Such risks, however, did not deter people from turning to online sex work. Indeed, Jones (2020) claimed that the internet has contributed to the emergence of new opportunities—not only experienced sex workers, who shifted from offline work to online scenarios, but also for new users, attracted by the idea of pursuing a career in digital sex work. The interest in webcamming appeared to be underpinned by the likelihood of performing indirect sex work, avoiding the risks of sexually transmitted infection (STI), as well as the flexible working hours (Velthuis and Doorn 2020). Furthermore, it appears that digital sex work attracts the users due to the financial reward gained when performing online sex service. The idea of empowerment in relation to the pursuit of such a career has been reported by scholars such as Saulman (2021) who suggested that webcam models consider themselves emancipated by the possibility of negotiating fees, of choosing the type of sex performance, and of concluding offensive or inappropriate interactions.

The possibility of earning more money was reported to be one of the reasons that motivated people to turn into online sex work. For example, according to Jones' (2020) study, the desire for a decent income was the central motivation for most webcam models (e.g., education costs or childcare costs), although some other performers claimed that they turned to webcamming either to promote their offline sex service such as escorting or to pay off debts. For example, sex workers who experienced the declining business of mainstream pornography production alternatively provided services such as market phone sex, online sex or solo sex recordings, and selling products such as worn underwear (Döring 2020). In addition to the shift online by experienced sex workers, with the lowering of the market-entry barriers and the rapid development of cultural and economic mainstreaming of commercial sex (Brents and Sanders 2010), many newcomers with no previous experiences make the move into webcamming to supplement their incomes. This intensifies the competition and impacts on the revenue of the established cam models (Berg 2016). Such competition seems to have been growing over time and this is still continuing. The levels of market saturation persisted and increased with the outbreak of COVID-19,

affecting the work of established cam models and content creators on OnlyFans (Martinez Dy and Jayawarna 2020) a site on which performers share explicit content and also earn money by selling photographs and videos of themselves to viewers paying a monthly subscription fee to the site (Lines 2020).

### 1.3. Lockdown and the Rise of Adult Content Sites

The pandemic outbreak and associated lockdowns were followed by serious concerns for national economies, social interaction, and the health of human beings (Pascoal et al. 2021). Besides the increase in psychological disorders, such as depression and anxiety due to isolation and quarantine, the pandemic restrictions altered physical proximity and intimacy (Galea et al. 2020), and therefore individual sexual habits (Banerjee and Rao 2020). Banerjee and Rao (2020) reported that, during the lockdown, many people turned to digital platforms seeking explicit content due, in part, to sexual abstinence imposed on those not living with a partner. This is corroborated by Gillespie et al.'s (2021) reporting of statistics on growth in traffic on online adult websites such as Pornhub and other adult content sites during the coronavirus pandemic (e.g., Uzieblo and Prescott 2020).

Direct in-person sex work was particularly impacted by the pandemic, with severe disruptions in provision as well as purchasing of in-person services (Prior 2022). Although many governments introduced support schemes to ease financial strain experienced those unable to work due to the impact of the pandemic, sex workers are largely excluded from such schemes (Brouwers and Herrmann 2020). Online avenues of indirect sex work, which did not require physical touch, could offer an alternative for both providers as well as buyers. A decrease in amount of advertisements for direct sex work and increase in amount of advertisements for digitally mediated sex work were found during the initial stages of the pandemic in Italy (Cipolla 2020). In the UK, while some sex workers were able to move online during lockdown through provision of webcamming services and sales of self-made sexual content, others feared risking their anonymity or lacked resources necessary to do so (Brouwers and Herrmann 2020). Additionally, workers involved in different avenues of sex work, such as porn performers, also turned to indirect online sex work in order to account for loss of their usual income due to closures of production studios (Döring 2020). The findings of a study by Lykousas et al. (2020) revealed that during lockdown, a sharp increase in the number of new users was observed on FanCentro, a platform which allowed users to sell self-made adult content. Online sex work could therefore become a source of income for sex workers who lost their jobs during the pandemic, and perhaps also attract individuals who faced financial difficulties but did not carry out sex work prior to the outbreak.

### 1.4. OnlyFans

Self-made sexual content can be very profitable, and may require less effort than webcamming as once created it can generate income many times over (Sanders et al. 2018). Such content can be sold to customers via, for example, content delivery platforms which facilitate the transaction by providing required technology and financial services (Sanders et al. 2018). One such platform which gained popularity in recent years is OnlyFans, where users are required to pay a monthly subscription fee in order to gain access to material uploaded by 'creators' (Ryan 2019). Although OnlyFans is not marketed as a platform for sexual content, many individuals use this site to sell self-made explicit pictures and videos (Ryan 2019). The platform also provides the creators with live streaming options (Ryan 2019). While OnlyFans takes a percentage of sales the commission fee of 20% is significantly lower than that of other webcam platforms, which may take more than 50% of the model's earnings (Ryan 2019; Sanders et al. 2018). OnlyFans experienced a 40% increase in performers during March 2020 at the beginning of the first lockdown (Martinez Dy and Jayawarna 2020).

*1.5. Aims*

Although the impact of COVID-19 in relation to sexuality (e.g., Delcea et al. 2021) and the consumption of pornography has been investigated in the literature (e.g., Zattoni et al. 2020), there is a lack of qualitative research into the effects of the pandemic on online sex work and its subsequent mediatisation. The aim of the current study is to fill this gap in the literature by analyzing the reportage in news media in the United Kingdom during the COVID-19 pandemic. Specifically, it explores the potential changes related to the saturation of the sex industry and investigates whether the market was reported to have increased, both in terms of performers and viewers, during the pandemic. Moreover, the study investigates how the coronavirus pandemic was reported to have affected the lifestyle of digital sex workers, with a focus on the possible variations in income levels, routines, attitudes, and requests for services.

The mainstream news media, whether accessed online or in hard copy format, is an important source of public opinion. During the pandemic in particular, people turned to mainstream media as other incidental sources such as work or social networks were curtailed. Given the links between mainstream media sources and social media, through the accounts of the corporations, and/or individual journalists driving traffic to their main site, narratives and counter-narratives developed in news media are a powerful source of public information.

A systematic analysis of press coverage is highly relevant, as it is a place where the public not only look for information on the current events but such outlets are crucial when it comes to the construction of topics like online sex work. As press articles have the power to affect peoples' perception of the world around them, it is important to attend to the ways in which digital sex workers' experiences are represented during the pandemic. Additionally, press articles can provide a space where their voices can be heard with an immediacy that hindsight research cannot. It is therefore an essential area to examine in relation to sex work trends, public perception of sex work, and its eventual destigmatisation.

## 2. Methodology

*2.1. Study Design*

The Grounded Theory (GT) approach (Strauss and Corbin 1990) was selected as an appropriate method to analyse the newspaper articles as this flexible yet systematic approach is frequently used in exploratory research. Its inductive techniques allow researchers to develop concepts of previously unexplored areas of research, and this in turn enables the production of a theoretical model that is strongly grounded in the data. The initial set of newspaper articles was subjected to theoretical sampling. Although the original focus was on the impact of the pandemic on the webcamming industry, it became evident with theoretical sampling that many of the press articles also covered additional forms of digitally mediated sex work, with particular focus on self-made sexual content creation. This finding therefore guided the subsequent sampling, which consequently allowed us to develop a theory with regards to the perceived impact of COVID-19 on online sex work more broadly. In accordance with Grounded Theory methods, the collected articles were analysed line by line and open coded. Next, the data were broken down further during axial coding, which allowed us to identify its categories, properties and dimensions. Lastly, selective coding was used to integrate the emergent categories, which consequently enabled the identification of the core categories, sub-categories, properties and dimensions in the development of the final theoretical model.

*2.2. Procedure*

The data searches were conducted between January 2021 and May 2021 using two search engines: Google and ProQuest-European Newsstream. The following keywords were used to identify newspaper articles published in the United Kingdom during the pandemic: camming; webcam models; cam girls; cam; webcam sex labour; webcam women; cam models; webcam sex work; cam boys; and webcamming. During the investigation, it was clear that the research question needed to expand to consider a wider range of the

digital sex industry. The keywords were therefore expanded to include digital sex work; sex workers; online sex work; digital sex; digital sex labour; content creators; digital sex performers; online explicit content; adult websites; online sex performance; online sex industry; and digital sex industry. These keywords were used either as a single search parameter that included a list of newspaper articles published in the period going from the 1 January 2020 to the 31 May 2021 or as an addition to keywords related to the pandemic, namely COVID-19; Lockdown; Covid; Pandemic; Coronavirus; quarantine; self-isolation; and 'social distancing'.

*2.3. Materials*

In order to be selected for inclusion in the study, the content of the news sources had to refer to the effects of the coronavirus pandemic on the digital sex industry, specifically providing information on the potential changes occurring within the online sex market amid the COVID-19 pandemic. The researchers sourced a wide range of media and a total of 40 articles were sourced from 19 different media sources. Of those, 20 articles were obtained from tabloid and regional newspapers (Mirror, Daily Star, The Sun, The Standard, The Metro, The Scottish Sun, Daily Record, The Sunderland Echo, Birmingham Mail), 15 were from 'broadsheet' style newspapers and periodicals (The Guardian, The Independent, New Statesman, The i, The Spectator, Wales on Sunday, The Times, and The Telegraph) and five were obtained from broadcast news (BBC, Sky News).

The 40 articles were divided equally into a set of 20 articles each for each of two researchers to analyse independently of each other (author A and author B) in Study One and Study Two, respectively. The division of the articles between the two researchers was done using two criteria: the balance of the political orientation of the newspapers (an even distribution of centrist, right- and left-wing press for each researcher) as well as a balance of the sources (an even distribution of broadsheet, broadcast media and popular journalism articles for each researcher). This enabled, as far as possible, two similar datasets for a a level of inter-rater reliability and thus a more robust analysis. A list of the articles analysed in each study, together with publication date, political orientation and number of sex workers interviewed can be found in Table 1 for Study One and in Table 2 for Study Two. In this way, the procedure follows that of Frost et al. (2010) and utilises the impact of different researchers on parallel data sets to add to the literature on pluralistic and mixed methods in feminist sex work research.

**Table 1.** Articles analysed in Study One, Online sex work from January 2020 to May 2021, Publication dates, Political orientation and Workers interviewed.

| Source | Published on | Political Orientation | Online Sex Workers Interviewed |
|---|---|---|---|
| **Broadcast Media** | | | |
| *BBC* Coronavirus: How do sex workers keep safe during pandemic? | 19 May 2020 | Centrist | One Woman (Unknown name) |
| *BBC* OnlyFans: 'I started selling sexy photos online after losing my job' | 15 July 2020 | Centrist | Mark Rebecca Lexi |
| **Broadsheet and Periodical** | | | |
| *The Times* Meet the king of homemade porn—a banker's son making millions | 26 July 2020 | Centre-right | Rose Jones One Woman (Unknown name) Kita Chidwick Soph Ana |

**Table 1.** *Cont.*

| Source | Published on | Political Orientation | Online Sex Workers Interviewed |
|---|---|---|---|
| *Telegraph*<br>OnlyFans: why are A-list celebrities joining a pornographic site? | 29 August 2020 | Centre-right | No sex worker interview |
| *New Statesman*<br>By exploiting loneliness, OnlyFans became the porn industry's great lockdown winner. But at what cost? | 4 November 2020 | Centre-left | Lisa<br>Janine<br>Alice |
| *The Guardian*<br>'Everyone and their mum is on it': OnlyFans booms in popularity during the pandemic | 22 December 2020 | Centre-left | Jah Bella<br>Zahra Sardust<br>Brooklin Rose<br>Rosewarne<br>Avalon Fae |
| *The Independent*<br>Pandemic makes prostitution taboo in Nevada's legal brothels | 20 February 2021 | Centrist | Alice Little<br>Allissa Starr |
| *Wales On Sunday*<br>Young Welsh scientist says posting explicit content online during pandemic has helped to pay off debts | 10 December 2020 | Populist | Rachel<br>Cariad |
| **Regional** | | | |
| *Metro*<br>What it's like to be a sex worker in lockdown | 22 May 2022 | Centrist | One Woman (Unknown name) |
| *Metro*<br>Cam girls are working inside plastic pods during lockdown | 23 August 2020 | Centrist | No sex worker interview |
| *Birmingham Mail*<br>'I can't get away from my wife'—Brummie love cheats turning to webcam girls in coronavirus lockdown | 5 April 2020 | Populist | One Woman (Unknown name) |
| **Tabloid** | | | |
| *Daily Star*<br>Coronavirus gives sex cam girls a business boom as randy customers stuck at home | 15 March 2020 | Non-political | Kate Kennedy<br>Joslyn Jane<br>Tyler Faith |
| *The Sun*<br>Sex workers demand aid from government because married clients 'can't sneak off' during coronavirus lockdown | 14 April 2020 | Right | One Woman (Unknown name) |
| *Daily Star*<br>Cam girls say they can earn more than £2000 a week—here's how it works | 22 April 2020 | Non-political | Elysia Downings<br>Scarlett<br>Lola Moon<br>Elizabeth |
| *Daily Star*<br>Older sex workers taught cam sex by younger women during coronavirus lockdown | 6 May 2020 | Non-political | Camila Hormazabal<br>Herminda Gonzalez Inostroza |

**Table 1.** *Cont.*

| Source | Published on | Political Orientation | Online Sex Workers Interviewed |
|---|---|---|---|
| *Daily Record*<br>Scots women warned sex webcamming is not easy way to make money during Lockdown | 8 May 2020 | Centre-left | Chelsea Ferguson<br>Megan Furie |
| *Mirror*<br>'I lost my job during lockdown so became a cam girl to pay the bills' | 5 May 2020 | Centre-left | Roxie |
| *Daily Star*<br>Cam girl raves about new sex trend that makes 'imaginations run wild' | 19 August 2020 | Non-political | Corrina Wild |
| *The Scottish Sun*<br>'I swapped my office job for online sex work in lockdown . . . but my fiance loves it' | 6 December 2020 | Right | Emma |
| *The Sun*<br>Horny men with secret covid fetish pay me £4k a month to cough and sneeze on FaceTime, reveals Page 3 girl | 6 May 2021 | Right | Holly McGuire<br>Melissa Howe<br>Kate McGrew |

**Table 2.** Articles analysed in Study Two, Online sex work from January 2020 to May 2021, Publication dates, Political orientation and Workers interviewed.

| Source | Published on | Political Orientation | Online Sex Workers Interviewed |
|---|---|---|---|
| **Broadcast** | | | |
| *BBC*<br>Coronavirus: Offine sex workers forced to start again online | 7 April 2020 | Centrist | Goddess Cleo<br>Eva de Vil<br>Gracey<br>Lizzy |
| *BBC*<br>OnlyFans: Digital footprint concern raised over images | 9 December 2020 | Centrist | Cariad<br>Rachel |
| *Sky News*<br>Coronavirus: More students are turning to sex work during COVID-19 pandemic | 10 September 2020 | Centre-right | Rose |
| **Broadsheet and Periodical** | | | |
| *The Guardian*<br>'The sex industry is not pandemic proof': workers in Australia faced with impossible choices | 7 November 2020 | Centre left | Jenna Love<br>Sarah |
| *The Independent*<br>The sex workers working from home | 19 April 2020 | Centrist | Mileena Kane<br>Allie Awesome<br>Remi Ferdinand<br>Betsy<br>Raie<br>Cecilia Morrell<br>Valentine |

**Table 2.** *Cont.*

| Source | Published on | Political Orientation | Online Sex Workers Interviewed |
|---|---|---|---|
| *The Spectator* There's nothing 'empowering' about the sex work on OnlyFans | 15 April 2020 | Right | Elodie Claudia Anon |
| *New Statesman* How the rich and famous stole OnlyFans from sex workers | 18 September 2020 | Centre-left | Abby Fairy Odelia Zelda |
| *The 'i'* Rise in online sex work 'linked to Covid poverty: SOCIETY | 4 December 2020 | Centrist | n/a |
| **Regional** | | | |
| *Evening Standard* OnlyFans is sex work and pornography–stop calling it 'empowering' | 11 September 2020 | Centre-right | Lisa Janine Alice anonymous man |
| *Metro* Cam girls are helping out lonely customers during the coronavirus lockdown | 2 April 2020 | Centrist | Red Delicious Your Goddess Jo Sofia Sanctuary |
| *Sunderland Echo* Heartbreaking stories of 'desperate' women who have turned to selling sex online to survive during pandemic | 4 December 2020 | Centrist | Anon 2 |
| **Tabloid** | | | |
| *Daily Star* Porn stars on coronavirus lockdown to swap sex scenes for solo Webcam streams | 20 March 2020 | Apolitical | n/a |
| *Daily Star* Cam girls fear they'll 'tum into sex robots as demand soars during lockdown | 21 April 2020 | Apolitical | Ava Moore Kat Aluna Chloe Sanchez Charlie Tantra |
| *Daily Star* Warning issued as increase in women turning to Webcam sex to make cash during lockdown | 8 May 2020 | Apolitical | Chelsea Ferguson |
| *Daily Star* Cam girl says 'men are gagging for it' more as coronavirus boosts business | 19 July 2020 | Apolitical | Lola Rose Curtis |
| *Mirror* Mum who lost job in lockdown set to become OnlyFans millionaire in just one year | 30 March 202l | Centre-left | Gracey Kay |
| *Mirror* Desperate students tum to sex and OnlyFans after lockdown kills pub and shop jobs | 10 April 2021 | Centre-left | Sophie McBumie Ind Aalyiah |
| *The Sun* Hooker move sex biz online: Escorts use web to service clients. Irish escorts start working from home. | 19 April 2020 | Right | n/a |

**Table 2.** *Cont.*

| Source | Published on | Political Orientation | Online Sex Workers Interviewed |
|---|---|---|---|
| *The Scottish Sun*<br>Nurse fed up with being 'taken for granted' during Covid crisis earns FIVE times her salary as an erotic cam girl | 1 February 2021 | Right | Cora Diamond |
| *The Scottish Sun*<br>'HAVE TO SURVIVE' Desperate students selling sex & signing up to OnlyFans as traditional bar jobs dry up and Covid forced shops to close | 1 April 2021 | Right | Aaliyah |

### 2.4. Ethics

Ethical approval was granted by the ethics committee of the School of Psychological Sciences, at Birkbeck, University of London, in line with the institution's protocols. Because the study involved the use of public information collected from newspaper reports, the researchers were able to investigate the area of interest without being intrusive into the workers' lives, interviewing them in person, or displacing them from their work activities at a time critical to their incomes.

## 3. Results

### 3.1. Study One

In the first study, a core category called 'Booming in lockdown: The digital sex industry' emerged along with three subcategories: 'Working as a Virtual', 'Beyond the sexual act', and 'I am in control and I am loving it!'. The first subcategory 'Working as a 'Virtual' comprised five properties which were: Established Virtuals; Virtuals overcoming the loss of previous non-sex related jobs', and 'Virtuals overcoming the loss of previously sex-related jobs'; 'The bursting competition' and 'incomes'. From the property of 'Incomes' the following dimensions emerged: 'more than before'; 'less than before'; 'enough for basic needs'; 'almost unchanged' and 'unsteady'. The subcategory of 'Beyond the Sexual Act' included the properties Chatting and Covid Fetish (see Figure 1). The subcategory of "I am in control and I am loving it!" included 'A Tool of Empowerment' as a property. This property comprised three the following dimensions: 'Choosing Working Hours', 'Choosing Clientele', and 'Setting Own Boundaries' (see Figure 1). The term 'Virtual' is an in-vivo extracted from one of the Daily Star, 6 May 2020: *"We call them 'the virtuals' and some can make a lot of money"*. The term indicates the range of sex workers working within the digital sex industry.

### 3.2. Booming in Lockdown: The Digital Sex Industry

The core category in Study One, 'Booming in Lockdown: the digital sex industry', encompassed the growth of the digital sex industry during lockdown and comprised both viewers and performers. The viewers were seeking entertainment and company to overcome the stressful circumstances, as reported by The New Statesman: *'The COVID-19 crisis has accelerated the commercialisation of sexual intimacy, providing temporary relief not only from sexual frustration but also loneliness . . . the online porn industry has grown even larger'* (New Statesman, 4 November 2020). The increase in performers resulted from many newcomers who, following unemployment, joined established online sex workers or following the closure of in-person workspaces, as suggested by the following extract: *'There has been a large spike in traffic to adult cam sites, along with a large influx of new models looking for ways to make money while remaining socially distant'* (Metro, 23 August 2020). The boom was experienced especially by performers working as cam models on webcam sites and also

models creating content for OnlyFans, and can be seen in the following extract: *'The business has been booming for webcam girls and OnlyFans models since the Covid outbreak sparked global lockdowns over a year ago . . . '* (The Sun, 6 May 2021).

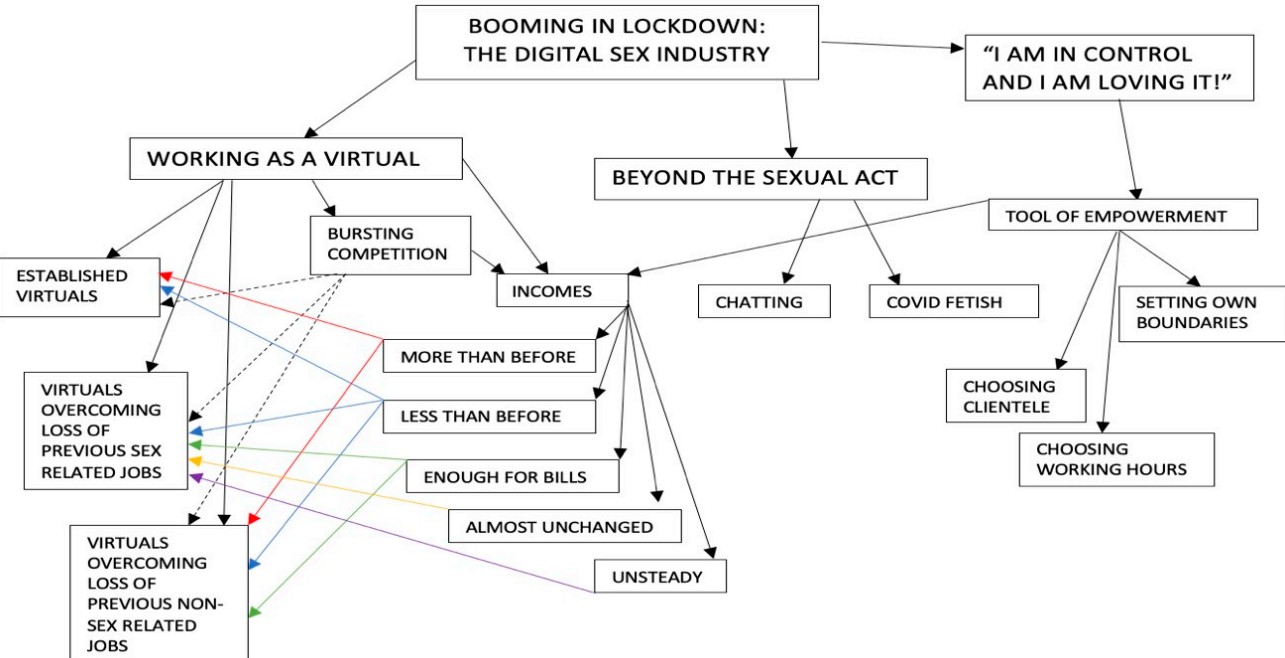

**Figure 1.** Diagram of the core category, subcategories, properties and dimensions.

### 3.3. Working as a 'Virtual' and Established 'Virtuals'

It was possible to identify from the data the first sub-category of 'Working as Virtual'. Three groups of sex workers were working as 'Virtuals' during the lockdown. With the onset of the pandemic and the consequent restrictions, online sex workers could continue their job as before with many women seeing an increase in their clientele. Cam girls were able to work from home and perform sexual acts for their paying viewers using platforms such as AdultWork and LiveJasmine, as stated in one of the analysed news reports: *'Cam girls and adult performers are enjoying a boom in business as quarantined customers make the most of their service'* (Daily Star, 15 March 2020).

Not all veterans were, however, able to keep working while staying home, as they were living with other family members. To provide an alternative workplace, one web company gave digital sex workers much needed equipment, as the Metro reported: " . . . *over in Columbia, an adult webcam company has taken things a step further by supplying its cam girls with plastic pods to work inside. These feature everything needed for the women to do their job; a laptop, bed or couch, and, of course, a webcam, with each room and the equipment cleaned and sanitised after use".* (Metro, 23 August 2020). This way, they could keep receiving income and providing for their families while respecting the pandemic restrictions.

### 3.4. 'Virtuals' Overcoming Loss of Previous Sex-Related Jobs

Among the women joining the digital sex industry, there was the property of established "offline" sex workers who previously performed sex services in a face-to-face fashion. The range included strippers and escorts, working in nightclubs and brothels, who turned to online services due to the social distancing and forced confinement of lockdown. The Birmingham Mail reported: *"The worker, who is camming instead of escorting as a way of paying her bills amid the coronavirus lockdown, said: 'I've had people who are talking to me but it's been late at night, like 1 am, and they want me to type to them, instead of talking to them because their wife is there"* (Birmingham Mail, 5 April 2020). Older sex workers who were not experts in the technology tools of media marketing sought assistance to advertise their sexual services

on digital platforms. Such women were taught by younger sex workers the required skills, as suggested by this extract: *"They're teaching others over WhatsApp how to get into it, how to find clients, how to set up an account to charge credit cards, how to sort a webcam. For the women over 45, it's not easy but you can always learn"* (Daily Star, 6 May 2020). Ergo, not everybody found the shift to digital easy.

### 3.5. 'Virtuals' Overcoming Loss of Previous Non-Sex Related Jobs

The second property was composed of those people who became unemployed with bills to be paid and more free time due to forced home confinement, and for whom work options were limited. This included those who were either furloughed or their posts made redundant or, in the worst-case scenario, dismissed. Losing their job led many people, of whom the majority were women, to turn into digital sex work with the advantage of working from home. Lockdown caused an increase in financial difficulties that was overcome by seeking alternative occupations, as suggested by one of the tabloids: *"The corona lockdown is hitting women's pockets hard, with 17 per cent of female employees working in shutdown sectors such as retail and hospitality. According to support groups, many women across Scotland faced with no income have turned to camming as a short-term source of income . . . "* (The Daily Record, 8 May 2020).

Some women were not only seeking an income to pay the bills, but they also took advantage of losing their previous jobs to pursue an option that would change their lifestyle: *"A woman known as Roxie (not her real name) was one such person who lost her job during this difficult time . . . Roxie was working in marketing, but soon realised that if she wanted to keep paying for her bills she would need to find a different way to make money while in quarantine. She wasn't too worried about this as she didn't really love her marketing role to begin with, . . . the 27-year-old had always been interested in pursuing a career in the sex industry . . . "* (Mirror, 5 May 2020).

Unsurprisingly, almost all the collected data gave a gendered representation of the financial issues experienced by women and without considering the role of men turning into online sex work to overcome the loss of income in the pandemic. Only one data source in our analysis could be found to the contrary in the following scenario: *"It was through necessity, I needed an income. It wasn't because I wanted to just get naked or post pictures of myself"* says Mark. He lost his job because of coronavirus in March and began posting semi-nude images on a subscriber-based social network. *"I applied for every single job I could find–all of the supermarkets, anything that was on the JobCentrewebsite–I applied for them all."* (BBC, 15 July 2020). In this case, the man turned to online sex work, not as a voluntary choice triggered by the idea of changing lifestyle, but rather he was pushed by lack of income to pay for essential needs.

### 3.6. Busting Competition in a Saturated Market

With the upsurge of women from different backgrounds working as 'virtuals', the digital sex industry experienced an oversaturation of the market. Such overload led to the property we identified relating to increased competition of 'virtuals' on different adult camsites and on *OnlyFans* in particular. Those in the digital sex industry faced intense competition as reported in The Guardian: *"Once the pandemic came around and strip clubs started shutting OnlyFans became so oversaturated because it was just every single sex worker, or just anyone in general, realizing there was money to be made"* (The Guardian, 22 December 2020). The addition, therefore, of newcomers to an established virtuals' market created a perception of an oversaturation of the digital sex market, as this excerpt shows: *"There has been a large spike in traffic to adult cam sites, along with a large influx of new models looking for ways to make money while remaining socially distant"* (Metro, 23 August 2020).

The growing competition, however, impacted the financial profits of all the virtuals and one news outlet reported: *" . . . she said that relying on income from the site is precarious as more users join and fuel competition."* (Wales on Sunday, 10 December 2020). Based on the data, the oversaturation of the market could be considered one of the factors linked to the

altered financial incomes but it may not have impacted all the adult digital platforms in the same way.

### 3.7. Continuum of Earned Income Levels

Despite the assumption that the pandemic may have augmented financial resources for 'virtuals', a variance could be detected in the analysed articles as to the income levels earned and this was not always represented as increasing. It seems that some digital sex workers (both experienced and inexperienced) struggled financially during this period. The property of earned income was dimensionalised in the following way: more than before, less than before, enough for basic needs, almost unchanged, and unsteady. This continuum is explained below.

### 3.8. More than Before

For the majority of women who turned to online sex work for the first time, the incomes earned were greater than their previous earnings. The majority of newcomers who claimed to be earning more than before seemed to be the ones who were reported to enjoy the job the most and had sought a change in their lifestyle having previously been working in non-sex-related occupations. In one report, for example, Emma, who swapped her office job for digital sex work during the pandemic, declared an increase in her salary, which had tripled compared to her previous monthly pay: *"In the first weeks of camming full-time I earned the same amount I would have earned in three months in my office job–about £3k"* (The Scottish Sun, 6 December 2020). By contrast, only a few established 'virtuals' reported an increase in their incomes during the lockdown. For example, the Daily Star reported the following about an OnlyFans content creator: *"Joslyn Jane from Miami is another OnlyFans star who sells sexy solo videos and "partner performances". She says she's currently making about $1700 (£1385) a week, up from her previous average weekly earnings of $1050 (£855)"* (Daily Star, 15 March 2020. Interestingly, the data did not show previous 'offline' sex workers turning to digital who reported an increase in their incomes. Hence, not all the 'virtuals' were as fortunate in terms of financial gain from digital sex services.

### 3.9. Less Than Before

Some sources reported that digital sex workers, both established and novel, experienced a decrease compared to their pre-pandemic income. Women already working as 'virtuals' claimed that, compared to the period preceding the lockdown, they had a less clients as a consequence of the forced confinement and co-existence with other members of the family: *"Virtual sex was expected to boom, but webcam girls and sex chat hostesses say their work has also dried up. One sex chat hostess said: 'Most of my regulars are married blokes with nowhere to sneak off at the moment. There is no way they will be calling while they are stuck in the house with wives'"* (The Sun, 14 April 2020). A similar outcome was reported by sex workers who previously worked as offline performers and lost their job due to the restrictions requiring social distancing regulations, as one stated in The Independent: *Legal brothels have been shuttered for nearly a year, leaving sex workers to offer less lucrative alternatives like online dates* . . . (The Independent, 20 February 2021). In another example, a nightclub sex worker, Camila, reported in one of the tabloids that her calls would provide financial gain but the income were barely comparable to her previous earnings in 'offline' sex work: *"The calls bring in something, obviously it's not the same and the money is not even what it would have been on a bad day before but it's something"* (Daily Star, 6 May 2020).

### 3.10. Enough for the Bills

Several of the extracts contained reports that newcomers would earn enough to pay for the bills, regardless of their previous occupation. Potentially, such an outcome could be related to the fact that turning online and create content for a wage was not accompanied by enjoyment but rather by the need to 'survive'. In one broadsheet article interviewing an offline sex worker who used to perform in one of the Nevada brothels, they were reported

to say: *'At this point, I am able to survive. I am able to pay my bills. I'm able to put food on the table, but I have had to dip into my savings'* (The Independent, 20 February 2021). Similarly, another broadsheet article, this time related to a 'virtual' who lost a non-sex-related job, clearly stated that incomings of digital sex workers are not always leading to the expected financial lifechanging: *'Now she works about six hours a day taking lingerie photos and marketing herself on social media. In the past month she has earned £170–about £6 a day. "It's harder than I imagined it would be," she said. "It's not as easy as people on the internet make it seem". The money she makes is what she needs to live'* (The Times, 26 July 2020).

Only a few of the news outlets reported the incomes of the established 'virtuals' to be similar to before the pandemic, as one article stated: *"Cam girls are struggling to keep up with the increased client load of lockdown, but even before the increased traffic, they were pulling in big bucks"* (Daily Star, 22 April 2020). Others also reported the precarity of the work as related in The Guardian: *'It was about a year before I really started seeing any type of income and even then it's not like a steady incomes or anything' she says. "I can have some days where (I) only make maybe $10, and then (I) can have weeks where (I've) actually cashed in about $500."* (The Guardian, 22 December 2020).

The above findings suggested a divergence in the way 'virtuals' perceive their incomes and that their perception relates to their income and type of work prior to the pandemic. Those not in sex-related occupations reported an increase but those doing in-person sex work previously saw a drop in income as the online market became more saturated during lockdown. The findings suggest that turning to online sex work was a time-consuming occupation and it did not meet the financial expectations assumed at the outset if someone was used to an in-person sex-related income.

### 3.11. Beyond the Sexual Act
Chat, Companionship, the 'Girlfriend Experience', and the Illusion of Intimacy

The clients' requests were reported as extending beyond the performance of sexual acts and this constituted a further sub-category of the data. Because of imposed restrictions such as compulsory social distancing, forced quarantine, and isolation to contain the spread of the virus, people started to seek company and alternative entertainment to overcome adverse feelings, as reported by the following extract: *'The COVID-19 crisis has accelerated the commercialisation of sexual intimacy, providing temporary relief not only from sexual frustration, but also loneliness'* (New Statesman, 4 November 2020). Many clients sought to chat or have a conversation with the 'virtuals' and many interactions with clients were based on conversations unrelated to sexual performance but rather centered on human companionship. A majority of sources reported that 'chatting' was a service offered to regular clientele as described in the New Statesman: *'OnlyFans offer what should best be understood as the "girlfriend experience" of porn. Successful creators sell not just explicit content, but also the impression of authentic personality. Creators are expected to message users privately, and perhaps remember their birthdays, or their children's names, thus offering the illusion of intimacy'* (New Statesman, 4 November 2020). Beyond this classic example of emotional labour, some 'virtuals' went beyond engaging in conversation during the digital paid time and were reported to be checking on their regulars' mental and physical health, as stated in: *" … during lockdown I have been texting some of them regularly to see how they are doing and if they have someone who is able to get groceries for them. They are too far away from me to help, but it's important for me to know that they are OK, and it gives them someone to talk to for a bit … "* (Metro, 22 May 2020). This was echoed in The Telegraph, which reported: *"Successful creators sell not just explicit content, but also the impression of authentic personality"* (The Telegraph, 29 August 2020)

### 3.12. Covid Fetish

Some clients sought more unusual trends to entertain themselves and from the data emerged previously unknown kinks. In some of the reports, it was stated that clients were requesting a virtual service directly related to the pandemic that was identified as a Covid

fetish, as described in this excerpt: " ... *she has regular punters who ask her to sneeze and cough on camera for up to 20 minutes at a time or even ask her to wear a face mask throughout their steamy calls.*" (The Sun, 6 May 2021). So, it can be seen that there was a continuum of requests from chatting to Covid fetishes that went beyond any sexual acts.

### 3.13. 'I Am in Control and I Am Loving It!'

While financial need was the main reason to turn to such a challenging industry, other reasons emerged such as the pursuit of a different way of life that afforded a higher income and more control. And in the category that emerged from the data in the final sub-category in Study One, i.e., the virtual sex workers' perception of being in control of their job and loving that control. It appeared that the virtuals who were achieving great financial gain were the ones who enjoyed the lifestyle provided by virtual sex work and took their time to create high-level content for their clients.

### 3.14. Virtual Work as a Tool of Empowerment

Many women were reported as perceiving their digital sex work to be a 'tool of empowerment". This meant not only being economically independent, but the increasing self-confidence 'virtuals' gained by setting their own boundaries, scheduling their work hours, and choosing their clientele. The following excerpt describes how one woman benefitted from the new work lifestyle: *"And for Jah Bella it's been life changing. 'I have a history of anxiety and depression and I have a lot more freedom to live life,' she says. 'I control my work hours. I don't need to feel guilty for staying home ... if a man abuses me or harasses me [online] I can just block them instead of having to deal with them every day.'"* (The Guardian, 22 December 2020). Additionally, some women were reported as relishing being able to set their own boundaries, and being able to control and choose the genre of performance, as in: *"I only go topless, and I'm in control of what I do, so I don't have to do anything I don't want ... "* (The Scottish Sun, 6 December 2020). Others who started working as digital sex workers when in urgent need of an income reported that they would likely continue the work: *"I wouldn't have made it if it wasn't for the pandemic, but now I don't know when I'll stop."* (The Times, 26 July 2020).

The notion of empowerment was characterised by the ability to set their own boundaries, be in control of one's working hours, choice of client, and the type of content uploaded onto the site. It was a complex balance, as reported by the following excerpt: *"Women have told us clients are becoming more demanding and financial pressure is making women feel they can't set boundaries they would want to"* (Daily Record, 8 May 2020). The feeling of empowerment was thus disrupted if the main reason for joining was the necessity of incomes, and if they felt pressured to shift boundaries.

### 3.15. Study Two

The core category which emerged from the data in Study Two was that of 'Increased engagement' which was was observed in three areas: the sites involved in provision of online sex services; new digital sex workers joining the industry; and those who were involved in provision of online sex services prior to the pandemic. These sub-categories were called 'sites', 'newcomers' and 'established online sex workers', respectively. The sub-category 'sites' consisted of one property: 'the booming business', whereas the sub-categories 'newcomers' and 'established online sex workers' each comprised three properties. For 'newcomers', these properties comprised 'reasons for joining online sex industry', 'perspectives of online sex work' as well as 'transition from direct/indirect sex work to online sex work'. For the sub-category 'established online sex workers', the properties included 'increased workload', 'increased competition' as well as 'increased desire for intimacy and companionship". The relationship between these concepts is illustrated in Figure 2 below:

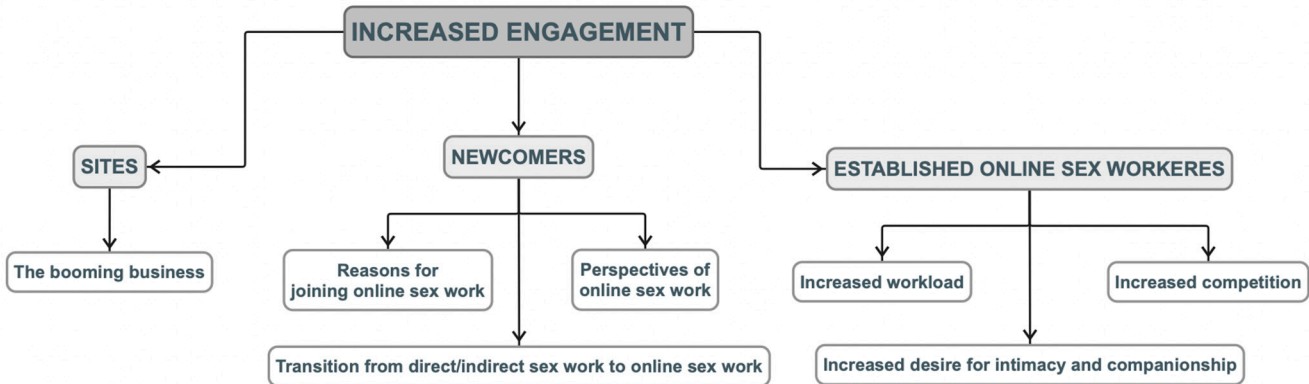

**Figure 2.** Diagram of core category 'increased engagement', sub-categories, and properties.

*3.16. The Booming Business*

An influx of both new subscribers and content creators on the various online sexual content sites was evident in the majority of the newspaper articles. This increase was noted in statements such as *"The coronavirus pandemic has driven a huge increase in traffic to webcam sex sites"* (Daily Star, 8 May 2020). Some articles provided specific evidence to quantify the change which occurred during the pandemic, for example, in: *'At CamSoda, the number of new viewers to the site has doubled this year when compared with early 2019, according to the company.'* (Independent, 19 April 2020). Other news outlets noted specific increases in sales: *"The website has seen a 15% spike in video sales since the outbreak of coronavirus and a 'steady rise' in the use of its streaming service."* (Daily Star, 20 March 2020). In September 2020, New Statesman reported: *"Like many other sites, the growth of OnlyFans was sent into overdrive by lockdown. As its popularity surged—traffic to the site more than doubled between March and August—it received prominent media coverage from BuzzFeed to the BBC. By May, OnlyFans had racked up more than 24 million registered users."* (New Statesman, 18 September 2020).

In December 2020, BBC announced the most recent statistics with regards to OnlyFans' explosion in popularity during the pandemic: *"The site claims users have risen from 7.5 million in November 2019 to 85 million–with 750,000 so-called content creators."* (BBC, 9 December 2020). A continuous growth in the number of subscribers as well as content creators was reported by OnlyFans throughout the pandemic. In April 2020, an article in The Independent stated: *"OnlyFans (…) reports a 75 per cent increase in overall new sign-ups –3.7 million new sign-ups this past month, with 60,000 of them being new creators."* (Independent, 19 April 2020). This increase in number of webcam models joining camming sites was reported for other sites including CamSoda, ManyVids as well as Chaturbate as seen in this excerpt from The Independent: *"Daryn Parker, the vice-president of CamSoda, says there has been a 37 per cent increase in new model sign-ups this March, compared with last March. For the same period, Bella French, co-founder and CEO of ManyVids, another camming site, says that there was a 69 per cent increase in new model sign-ups."* (Independent, 19 April 2020). This reportage was corroborated by the BBC in: *"US-based livestreaming site Chaturbate has reported a 75% rise in the number of sex workers signing up since the outbreak began–an increase faster than the rate at which audience traffic is rising."* (BBC, 7 April 2020).

*3.17. Newcomers*

The next sub-category was that of newcomers, which was divided into three properties, the first of which was 'reasons for joining online sex work'.

*3.18. Reasons for Joining Online Sex Work*

The majority of articles cited the main reason for entering online sex work as the need for an alternative income, often following loss of work due to the lockdown. For some women, particularly those who were mothers, the decision appears to be fuelled by desperation, caused by inability to pay bills and provide for their families, as in: *'Gracey*

*found herself unable to work when the UK was plunged into lockdown in March last year. She found herself with no cash and the bills were mounting up so she turned to OnlyFans after a friend suggested it to her. "I was panicking," she said. "I had all these bills to pay and no income. It was stressful'* (The Mirror, 30 March 2021).

Although the pursuit of income seemed to be a primary motivation for the majority of those who became involved in online sex work in the midst of the pandemic, others had secondary motivations. For Rose, a student, turning to digital sex work was motivated by both the loss of her job in hospitality and her health concerns related to the virus: *"Rose, which is not her real name, turned to online sex work after she effectively lost her job in a restaurant as a result of lockdown. She says she is also classed as high-risk in relation to COVID-19, so working through a subscription website was "safer" and "financially easier"."* (Sky News, 10 September 2020). In another media outlet, Pea, also a student, found digital sex work offered source of entertainment, convenience, as well as freedom in addition to an income. She was reported to say: *"I started at the beginning of lockdown because I was bored and skint. (...) it fits in nicely with uni work. It's just a part-time job with a difference. I can just do little photoshoots whenever I want."* (Mirror, 11 April 2021). Such decisions were echoed in the regional press reportage, with decisions about online sex work motivated by financial gain, as well as companionship and improving self-esteem: *"[Lockdown has mean] it has been very hard finding jobs or even coping with hardly no support. I turned to [advertising to sell sex] because I got the cash to help me and the attention that people wanted me as I was very lonely."* (Sunderland Echo, 4 December 2020).[1]

While the bulk of narratives in the articles revolved around loss of job or inability to work due to the effects of lockdown, there were women newcomers to the work who had made a conscious decision to quit their jobs in the midst of the pandemic and digital sex services became their main source of income. This choice appears to be primarily motivated by lack of previous job satisfaction, with financial motivation being secondary. For Rachel, an ex-data analyst, it was the pursuit of a *'better work-life balance'* (BBC, 9 December 2020). For Cora, an ex-nurse who was previously involved in online sex work in her free time, it was an opportunity to follow her *'dream job'* (The Scottish Sun, 11 February 2021) as a full-time webcam model.

*3.19. Perspectives of Online Sex Work*

Newcomers' perspectives on online sex work are presented positively as well as negatively. Aspects of online sex work highlighted by the newcomers in the articles included the perks, which for many included significant financial gain, often achieved in a short period of time. The actual income generated through provision of online sex services differed between the women, ranging from *'£400 a month'* (Mirror, 11 April 2021) to *'more than £2,800 in four weeks since launching (the) page'* (BBC, 9 December 2020). Additionally, newcomers to online sex work often made much more money in their new job than in their old one. According to Cora, an ex-nurse: *"In front of the camera I could earn three to five times more."* (The Scottish Sun, 11th February 2021). An extreme example is Gracey, who made *"more than twice her annual £18,000 hairdresser income in one month."* (Mirror, 30 March 2021).

In addition to reports of financial gain, the advantages of the provision of digital sex services for some women included increased confidence. In many reports, women had no regret and even expressed being liberated by their new job, claiming a sense of empowerment and freedom. This was particularly striking in the account on Sky News: *'The woman, in her 20s, says she "loves" her new source of income and refers to it as "one of the best things" she has ever done. "I used to be anorexic and now I am the most body confident I have ever been in my life. (...) (It's) very, very liberating. I'm in control of my own content and of my own body and I can do whatever I want with it because you can set your own levels'* (Sky News, 10 September 2020).

While for some women online sex work was perceived as safe (or mostly safe), for others it presented risks, particularly of being recognised by someone personally known to them as well as exposure to criticism. Some women therefore decided to keep their new job a secret from their family members, perhaps in an effort to avoid disapproval as

expressed in this report: *'"I think the main risk to me was in going public, you're opening it to people you may know in real life and not everyone is going to agree," she said. "You have to be strong enough to know that you more than likely will come across criticism, it's not as easy to brush off for everyone, nor is it easy to defend yourself."'* (BBC, 9 December 2020). Criticism from relatives and acquaintances was not the only fear as online sex workers may be targeted by individuals for whom their mere existence is problematic. This was seen in the reportage in The Mirror: *'Some people, and it's a lot of men, have said horrible things online. Some people have even subscribed to my page, and paid to do so, just to tell me they don't like what I'm doing'* (Mirror, 30 March 2021). Similarly, one newcomer reported that it was a lack of support which prompted her to engage in online sex services in the first place: *"It did make me feel dirty, that I had to turn to this as it shouldn't be that way. People should have more support, I was reaching for help but no one was helping me"* (Sunderland Echo, 4 December 2020). This exemplifies the notion that online sex work was the only financial safety net for some at the beginning of lockdown.

### 3.20. Transition from Direct/Indirect Sex Work to Online Sex Work

A rise in involvement in online sex work was also noted in reports of those who were already in sex-related services prior to the pandemic. The shared rationale for pivoting their work online was predominantly financially motivated, as sex workers were left unable to engage in direct in-person services following lockdown. For some, this switch was their first foray into digital sex work as this extract in The Independent shows: *'While working as a stripper in Oregon, Kelpie Heart had long thought about taking her work online. Then the coronavirus pandemic led to bar closures, and she found herself out of work. So, for the last month, Heart has begun streaming performances from home, doing one live show a week'* (The Independent, 19 April 2020). Others were already involved in online sex services in addition to their direct sex work; however, the majority of their income was generated though provision of direct sex services: *'"The virus is a disaster for client-facing businesses–and sex work is no different," says Goddess Cleo, a dominatrix from London. "Most of my income is generated from one-on-one sessions and events. I [normally] only make a bit of money through online avenues." But like many others, Cleo has switched focus to digital since the lockdown came into effect'* (BBC, 7 April 2020).

A worker's previous involvement in the sex industry appeared to facilitate the transition between direct contact sex work and the provision of online services. For many, however, the transition was much more complicated. Online sex work can require more patience from the sex worker and be more time consuming than one-on-one meetings. Furthermore, it may call for a somewhat different set of skills; marketing services is a necessity to achieve success online, as this extract showed: *'"It's not about flashing ya nipple and earning big bucks", wrote UK sex worker Gracey on Twitter. It takes ages to gain an online following and even longer for [them] to buy your content. (...) The marketing requires so much effort, it is unreal,'* (BBC, 7 April 2020).

Online sex work is often more public than sex work requiring direct contact with customer. As a result, the anonymity of many sex workers can become compromised, as the following extract shows: *"Those making money from sex work in person before the pandemic went on OnlyFans and realised you have to produce so much content to make money, and it's really hard to do that without including your face and voice in videos."* (The Scottish Sun, 11 April 2020).

### 3.21. Established Online Sex Workers

The final sub-category in Study Two pertained to the experiences of established online sex workers during the COVID-19 pandemic. There were three properties of this subcategory the first of which was 'increased workload'.

### 3.22. Increased Workload

The rise in demand for online sex services, particularly at the beginning of lockdown was highlighted in numerous articles. The increase in clientele was observed by many

established online sex workers and often led to increase in income: *Ava Moore said that the lockdown had been positive for her business, as more people watched live shows and interacted on social media. (...) Her business has grown by 30% since the lockdown (...) March has been Ava's best month since 2017* (Daily Star, 21 April 2020). Increased income was not always a direct outcome of the increased demand, however, and some digital sex workers reported a much higher workload than before pandemic, but without the concomitant financial gain, as one report shows: *"I'm meeting a whole bunch of people more frequently than I normally would, but there's not much more money" Kane says'* (The Independent, 19 April 2020).

As the availability of online sex workers increased during lockdown for some it was an opportunity to invest more time and effort into the development of their business, as one report shows: *Allie Awesome, a cam model, works around 60 h a week, she says. Her work day begins right after waking up, when she looks through her social media notifications and checks in on her customers. (...) And as social distancing leaves her stuck inside, she has found herself working more than ever. Though she tends to work directly with individual customers through Discord, a chat app favoured by video game players, she's now using other platforms more, including Chaturbate, OnlyFans and Skype.* (Independent, 19 April 2020). While most online sex workers took advantage of the increased demand for their services, others were mindful of the need to balance this new increased demand with her own work life balance as this extract demonstrates: *'Kat Aluna said that she had more than 300 fans watch her live show this week. She said: "That's pretty big. But I don't necessarily want to ride this wave–I'm putting myself first, because I don't want to turn into a sex robot and stop listening to my body"'* (The Daily Star, 21 April 2020).

*3.23. Increased Competition and Marketing Trends*

Established online workers reported an increase in the number of newcomers joining the industry and expressed concerns at the competition this created, for example: *"Online dominatrix Eva de Vil says: "There's lots of new girls joining the scene right now–or offline sex workers moving online to help with finances." (...) UK sex worker Lizzy says camming has become even more competitive since the pandemic began* (BBC, 7 April 2020). The heightened competition due to increased numbers of digital workers meant that many established online workers had to work more hours or cut prices, and one report explained: *'Cariad said the increase in the number of people using the platform had led to more competition, meaning 'you definitely have to do more'* (BBC, 9 December 2020). In many cases, 'doing more' meant lowering prices with a hope of attracting more custom, one worker was reported thus: *'"I've seen quite a lot of girls running discounts," Eva says. "We're sensitive to [customers'] drop in income"'* (BBC, 7 April 2020).

Online sex workers reportedly noticed a change in customers' spending habits, specifically an apprehension to pay for services, as this extract states: *"Chloe Sanchez has found that customers now expect her content to be free, because sites like Pornhub have made access free since the lockdown began."* (Daily Star, 21 April 2020). In another report, clients were not paying as much: *"In Kane's experience, new viewers aren't tipping as much as they typically would."* (The Independent, 19 April 2020). Lastly, one digital sex worker noticed that her customers were seeking sexual release more strongly than before the pandemic and she also observed a change in the demographic of customers accessing her services. *"The model has also met a lot of men living alone. She explained in a Daily Star interview: "I'm not finding guys are more kinky, however they are certainly a lot more sexually frustrated. "I can tell they are gagging for it a lot more now. I'm also coming across a lot more married men in lockdown."* (Daily Star, 19 July 2020).

For others, however, increased competition meant increased marketing and more maintenance of their social media presence as well as the creation of special content, all of which led to increased workload, as the sex worker Ava said in this extract: *"Since the lockdown, I've been spending much more time on social media. I can show how available I am at the moment and tease my audience with special offers—a new morning live show and a Snapchat special for the lockdown. The Snapchat show lasts five minutes, costs [Euro]15 £13 and allows married men to wander off and pleasure themselves quickly."* (Daily Star, 21 April 2020). This shows

the reported marketing response by established workers who recognised quickly how to respond to the changing market for their services.

### *3.24. Customers' Increased Desire for Intimacy and Companionship*

Established online sex workers were reported as noticing a change in customers' requests and desires during the pandemic. This change often pertained to the type of relationship the customer sought with the worker. Several digital sex workers particularly noticed increased desire for intimacy and companionship, as in this report: *'I have never had so many cam-to-cam requests. Since this has all started, everyone wants cam-to-cam. It's as if there is that extra connection of face to face, which is something that we are getting less of in our day to day life under quarantine. For me, every single person has said, can we cam to cam, whereas before it was only 20–25%, but now everyone wants to do that'* (Metro, 2 April 2020). This was supported in another report: *'She said that private requests have also gone up by 30%. Chloe added: "The other day, two guys asked me to stay fully clothed–they just wanted to talk.'* (Daily Star, 21 April 2020), and another was reported thus: *'I've found that I've had an influx of silent phone chat requests more than anything.'* (Metro, 2 April 2020).

Online sex workers also commented how their interactions with clients reflected those in the rest of the community stating that COVID-19 had become the main focus of their conversations. One digital sex worker spoke about her experiences with new customers as well as key workers: *"'There are lots of new people I'm talking to which is great, so it's nice to have new conversations with people,' Red_Delicious adds. They're all working from home and the topic is always coronavirus, isolation and how everyone is in it together. It's interesting because it's almost become a focal point. (...) I've been speaking to a lot of frontline key workers (...) a lot of them have said how nice it is to talk to someone and how I brighten up their day"* (Metro, 2 April 2020). Another digital worker described her experiences with regular customers, expressing how going through difficult times together increased a sense of camaraderie between the workers and their clientele: *"'I think the loveliest part has been regular chatters coming in frequently to ask about my wellbeing and to make sure I'm ok (...) 'The coronavirus has been the main topic of conversation for a few weeks now and has made us all feel like we're a community again going through the same trials and tribulations sharing opinions and coping mechanisms. It's great knowing we are all helping each other through this time.'"* (Metro, 2 April 2020). This reflects a change in client needs to more personal, rather than sexual, intimacy with a need for silent companionship at the extreme end of the continuum.

## 4. Discussion

The findings from both studies show the reported impact of the COVID-19 pandemic on the digital sex industry. A trend of significant growth in users in webcamming in general and on *OnlyFans* in particular was reported across all media sources. The pattern of increased engagement in online sex work at the start of the pandemic was such that it was described as 'booming'. For the sex workers reported in the news articles, their experience of this boom depended on their previous occupation. For those moving into sex work for the first time, it represented an increase in earnings, for those already in sex work but moving online for the first time it represented a loss. Those already in online sex work noticed increased competition and had to pivot their marketing and services accordingly. This pivot took a variety of forms but included, for example, more social media marketing, or earlier and shorter videos as content creators took account of clients' working from home situation. The findings across the two studies showed the experience of newcomers as they transitioned to online sex work as well as the perceptions of experienced online workers who charted the change in client requests. These requests spanned silent companionship, to chatting, and more extreme requests and fetishes like the Covid fetish.

In Study One, it was possible to identify two groups of newcomers joining the digital sex industry and the established community of 'virtuals'. Those were either sex workers performing direct sex services, such as escorts, or new users who lost their jobs or wished to replace their previous employment with digital sex work. The market saturation led

to an increase in online competition among 'virtuals', which seem to be related to the changes in income earned during the pandemic. While the perceived revenue was not always as high as expected, perceptions varied depending on the type of jobs the 'virtuals' were performing prior to the shift into digital sex work. This finding is in line with Jones' (2020) results, suggesting that incomes are unequal among 'virtuals' but that in many circumstances, new users previously working in non-sex-related jobs were more likely to perceive their current incomes as better than their previous occupations.

A new concept that emerged from the study concerned the type of services performed by the 'virtuals', which did not involve sexual acts. Indeed, it was found that many of the 'virtuals' would be asked simply to chat with their clients and offer an authentic persona. The findings support Nayar's (2017) research suggesting that webcamming is not merely related to sexual encounters, since clients are often seeking companionship, hence someone to talk to. Furthermore, in line with Nayar's (2017) findings, providing authenticity is important for webcam models in order to meet clients' expectations. This finding supports Hall's (1995) formative findings of women on fantasy phone lines acting to gain patriarchal privileges of economic power.

The reports show that online sex workers perceived the job as empowering due to the ability to set their own working hours and choose their clientele, but also the ability to apply personal boundaries in relation to the type of service to be performed. Such empowerment was often linked to growing self-confidence and these findings are in line with Henry (2018), who reported that many 'virtuals' claimed that being a digital sex worker helped to develop and improve mental health issues and low self-esteem.

The concept of empowerment, however, was not shared by all 'virtuals', especially those who turned to digital sex work as temporary urgent alternative work to overcome financial hardship. Such women were often not able to set their own boundaries, for example, because of the economic distress that emerged amid the coronavirus pandemic. This supports previous findings, e.g., McCracken and Brooks-Gordon (2021) who illustrate the complexity of empowerment. Although many women feel empowered and safer doing online work than when delivering in-person services, due to the ability to turn off the computer or blocking the countries or viewers (Henry 2018), pursuing such a career may still be challenging. Those challenges may emerge, for example, when lacking the technological ability necessary to have a successful career. Velthuis and Doorn (2020) suggested that the job requires performers to have specific tech skills and present to the client an authentic persona that provides a high-quality interaction.

In Study Two, the findings of increased engagement develop the theoretical model further to show added nuance behind the increase in the number of models joining webcamming sites as well as increase in number of new viewers. Additionally, the expansion of OnlyFans in terms of subscribers and content creators was highlighted by the press throughout the pandemic. These findings correspond with those of Lykousas et al. (2020) but while their study focused on only one website, the current study provided novel findings of the increase in users observed across multiple platforms. Furthermore, the study explored reasons for entering online sex work in the midst of the pandemic and revealed that financial motivation was the most salient among new online sex workers. This was unsurprising as pre-pandemic research on motivations for entering the online sex market provides similar findings (Jones 2020). The ability to get work immediately, cited by many newcomers, is simply more marked in the pandemic experience. The analysis of the articles also revealed that many newcomers also had secondary motivations such as health concerns, boredom, convenience, loneliness and desire to increase one's own self-esteem. Such incentives appear novel and may be perhaps specific to the experience of the pandemic and social distancing, as the literature mostly highlighted motivations in the form of autonomy as well as experiencing pleasure and exploring one's own sexuality (Jones 2020). The findings also suggested that the COVID-19 pandemic presented an opportunity to change career paths for some new online sex workers who were unsatisfied with their previous jobs. This echoes the experience of other occupational groups during the

pandemic and perhaps their new employment will provide them with higher satisfaction, as pre-pandemic research suggests that individuals involved in provision of online sex services generally experience high levels of job satisfaction (Jones 2020).

Our analysis revealed that not only did many newcomers experience advantages such as significant financial gain, increased confidence, liberation and sense of empowerment but they also faced risks in the form of being outed as well as criticism and/or harassment. Such findings largely replicate those of pre-pandemic research on online sex work (Jones 2015, 2020; Sanders et al. 2018), in which both risks and benefits of online work were examined. Because the anonymity of those in sex work can be put into jeopardy by entering online avenues Brouwers and Herrmann (2020) suggested that this may be a factor which previously prevented many sex workers from involvement in the provision of online sex services. The current study did not find evidence that newcomers had been victims of online aggression such as doxing or capping (Jones 2015, 2020). These issues were not mentioned in the interviews or the articles and may well be an artifact of the journalist, rather than researcher, interviews.

The pandemic-driven transition from direct and indirect forms of sex work to the provision of online sex services was a new finding from the study. The analysis revealed that many sex workers turned to digitally mediated sex work in order to make up for the loss of income they experienced due to the effects of the pandemic, and these findings correspond with commentaries by Brouwers and Herrmann (2020) and Döring (2020). The transition from sex work requiring direct contact with customers to provision of online sex services was a complex one. The analysis revealed that indirect digital sex work could be more time consuming than other forms of sex work and marketing skills are essential to become successful and profitable. Similarly, evidence from the pre-pandemic literature suggests that building a brand and following in digital spaces is laborious, and it may take over a year to become prosperous (Jones 2020).

The analysis of the press coverage revealed that established online sex workers experienced an increase in demand for their services during the COVID-19 pandemic. This is a novel finding and adds to research into the effects of the outbreak of the virus which evidenced increases in other online areas, such as sales of sex toys and consumption of pornography (Arafat and Kar 2021; Zattoni et al. 2020). Furthermore, it has been found that many of the established digital sex workers increased their workload during lockdown in order to accommodate an increase in demand, with an exception of one worker who worried that increasing her workload could lead to negative physical and psychological consequences and set her work boundaries accordingly. The analysis of the press articles suggested that the majority of the established online sex workers invested more time into interaction with their customers as well as overall development of their business. Although for some of the online sex workers this led to increase in income, others found themselves working more but not earning any more than before the pandemic. This finding echoes pre-pandemic research in which high precarity of incomes in some forms of online sex work has been highlighted as an outcome of an inconsistent number of customers and working hours (Sanders et al. 2018, 2020). However, precarity of incomes may have worsened during the COVID-19 pandemic, as salaries of many consumers of online sex work were likely affected by the outbreak of the virus, which may have resulted in them becoming more frugal. This too is a novel finding and adds to the literature on the market economics of sex work (e.g., Brooks-Gordon et al. 2015; Cunningham and Kendall 2011) to increase our understanding of the economic nuances behind the phenomena observed.

Accordingly, the study explored the competition within the online sex industry during the COVID-19 pandemic as presented in the analysed articles. Although high competition in some areas of the online sex market has been highlighted in the pre-pandemic research (Jones 2020; Van Doorn and Velthuis 2018), the current study provides novel findings, suggesting that this industry was subjected to greater rivalry during the pandemic, which can be ascribed to the arrival of newcomers and increased availability of many digital sex workers. Moreover, the analysis of the press articles allowed us to identify some of

the strategies adopted by the established online sex workers in order to attract customers. These included lowering prices, increasing their workload by maintaining a regular social media presence, creating additional content, and different, shorter content in recognition of client time constraints. These strategies may not be exhaustive and should be investigated further, as pre-pandemic research argued that in order to stand out from the crowd, digital sex workers may engage provision of more extreme and sexually explicit content (Van Doorn and Velthuis 2018; Sanders et al. 2018; Jones 2020). Our research shows a more subtle modification to working practices.

Lastly, the analysis of the press articles enabled the exploration of the changes in customers' attitudes during the pandemic. The findings suggest that customers of established online sex workers experienced increased desire for intimacy and companionship, evidenced though increased requests for private interactions as well as interactions involving camera-to-camera and silent companionship in calls. Furthermore, many digital sex workers discussed the effects of the pandemic with their clientele, which increase bonding and created a sense of camaraderie. Although the importance of emotional labour in some forms of online sex work has been already highlighted in pre-pandemic research (Hall 1995; Sanders et al. 2020; Jones 2020), the findings of the current study suggest that the role of emotional labour in provision of digital sex services may be even greater during uncertain times and in extreme conditions, such as in the midst of a global pandemic. Additionally, changes in customers' spending habits, increased frugality of consumers of online sex services could be potentially explained by the global economic crisis caused by the pandemic. Moreover, one online sex worker noticed greater desire for sexual gratification amongst her customers, who increasingly included married men as well as men who live alone. Likewise, evidence from the literature suggests that lonely men were more likely to use sex as a coping strategy during the early stages of COVID-19 pandemic (Gillespie et al. 2021; Cocci et al. 2020) which could potentially explain the increased need for sexual release.

Besides companionship and chatting, it also emerged that viewers were requesting unusual performances, such as the Covid fetish in which sex workers were required to sneeze or cough on the camera. It has been argued that performing unusual services could have positive effects on the lives of performers and their clients, as it pushes sociocultural boundaries and allows moments of freedom from the 'regulatory forces in society' controlling individuals' sexual behaviour (Jones 2020). Therefore, the emergence of a Covid fetish may be perceived as a need to escape from some of the restrictions imposed amid the COVID-19, such as wearing a mask to avoid people from spreading germs through coughs and sneezes. This is a totally novel concept and a new finding that adds to the literature on sex work and on fetishes in particular.

### 4.1. Strengths

Newsmedia offers a valuable source for analysis with digital and hard copy editions as well as reach across social media, narratives and counter-narratives are forged in news media. They offer an immediacy that many 'hindsight' research interviews cannot. Researchers of sex work understandably make attributions about what is written in the press about sex work so it is important to analyse the press coverage itself, not least to prevent fundamental misattributions about what is said or written about in the news media—this is a critical strength of this study. The research did not seek to explore whether the coverage was positive or negative per se, but rather, by utitilising the immediacy of content made in real time, it sought to harness the content itself to show in a systematic way what that reportage contained. In doing so, this research provided many novel findings, outlining numerous ways in which online sex work was affected by a global pandemic.

A further strength was the use of two researchers, to use the potency of pluralistic, mixed methodologies (Frost et al. 2010) in the analysis of a parallel (but not identical) set of reports. The emergent Grounded Theory model provides real insight into the deployment of emotional labour, and the roles it plays, to add to the extant literature on emotional

labour (e.g., Hall 1995; Parvez 2006; Flubacher 2019) to expand our understanding of emotional labour in such contexts.

The current study is aware that the use of newspaper reports may limit the parameters of the information regarding topics related to digital sex work and, for example, the lived experiences of 'virtuals' may differ from those when engaging in an in-depth interview with a researcher experienced in sex work research, and it makes no claim for that. The study will provide a valuable comparator to in-depth interviews with sex workers so that their representation by researchers and journalists can be compared. The study does, however, reflect the growing awareness of the burgeoning industry in the mainstream press and the mediatisation of sex work.

### 4.2. Limitations

The findings of this study are also subjected to certain limitations. Firstly, as the study focused only on press articles released in the UK, and while the news media used data from online sex markets in a variety of other countries it is possible that the current findings are specific to the news media operating within the UK and in the English language. Future research could therefore investigate whether mediatisation of the digital sex industry is the same in other countries and other languages. It is important also to note the relative lack of representation of men and transgender sex workers involved in online sex services observed in the analysed press articles. Although women tend to predominate digital sex industry, an analysis of the experiences of men and transgender online sex workers could add depth to the current findings.

This study found only one report of a male worker, yet evidence of those involved in male sex work (McLean 2012; Morris 2021; Brooks-Gordon and Ebbitt 2021) is strong and found to be more 'incidental' in its inception. As suggested by Henry and Farvid (2017), it is important to consider the experiences of minority groups within the sex industry and future research could include experiences of male digital sex performers during the pandemic. Future investigations could therefore balance out the inherent cis/het bias in past findings.

Moreover, as the analysis in the current study was based on reportage obtained from newspaper coverage, the reliability of information can be compromised by varied journalistic standards and perspectives. Additionally, even though steps were taken to provide an overview of the effects of COVID-19 on many forms of online sex work, the analyzed articles centered around webcamming and sexual content creation, largely leaving out other forms of digital sex work such as phone sex chat or instant messaging—although these are not all mutually exclusive.

### 4.3. Future Research

While the findings of this study suggest that online sex work became more mainstream during the COVID-19 pandemic, it will be interesting to explore how this, like other patterns of home working, evolve in the future. The platform *OnlyFans* has been scarcely studied and yet it experienced a large spike in traffic during these times so it would be valuable to investigate whether the platform retained the high number of subscriptions or if performing on adult platforms decreased after pandemic restrictions were lifted. Such research would add to knowledge on the landscape and economics of sex work (e.g., Brooks-Gordon et al. 2015).

The sheer volume of media reports shows a degree of integration of the sex industry in mainstream culture and further analysis may also contribute to understanding how sex work could progress towards destigmatisation and increased public awareness. Additionally, given that the reported data have suggested clients' demands involving services going beyond the sexual act into basic companionship and also into illness fetishes, future research should investigate whether such demands assumed a different meaning for users (both performers and viewers) during these times, or whether the need for such intimacy and interaction was merely a response to the privations of the pandemic lockdown.

## 5. Conclusions

The mainstream news media is where most of the public get their information and these are trusted public platforms that are broadly inclusive. By exploring the reportage of the effects of the coronavirus pandemic on virtual sex workers engaging in webcamming and *OnlyFans*, our research is able to provide information on how some areas of the digital sex industry adapted to the outbreak of the coronavirus pandemic, increasing our understanding of the topic, and more crucially how the topic was reported and developed in a media-saturated society.

**Author Contributions:** Conceptualization: B.B.-G., V.R. and A.P.; Methodology: B.B.-G.; Analysis: V.R. and A.P.; Investigation: V.R. and A.P.; Original draft preparation: V.R. and A.P. did individual drafts, and B.B.-G. amalgamated them; Supervision: B.B.-G.; Project administration: B.B.-G. All authors have read and agreed to the published version of the manuscript.

**Funding:** This research received no external funding.

**Institutional Review Board Statement:** The study was conducted in accordance with the Declaration of Helsinki, and approved by the Ethics Committee of School of Psychology, Birkbeck, University of London, approval code 202066 on 4 February 2020.

**Informed Consent Statement:** Not applicable.

**Data Availability Statement:** All data is in the public domain and can be accessed via the sources in Appendix A.

**Conflicts of Interest:** The authors declare no conflict of interest.

## Appendix A. List of Source Newsmedia Articles

Abgarian, A. 2020. Cam girls are working inside plastic pods during lockdown. *Metro*, August 23. Available online: https://metro.co.uk/2020/08/23/cam-girls-have-abandone d-bedroom-are-workingwarehouse-pods-13166406/ (accessed on 4 March 2021).

Bakar, F. 2020. Cam girls are helping out lonely customers during the coronavirus lockdown. *Metro*, April 2. Available online: https://metro.co.uk/2020/04/02/cam-girls-s eeing-lonely-customers-coronavirus-lockdown-12479717/ (accessed on 4 March 2021).

Balloo, S. 2020. 'I can't get away from my wife'–Brummie love cheats turning to webcam girls in coronavirus lockdown. *Birmingham Mail*, April 5. Available online: https://www.birminghammail.co.uk/news/midlands-news/i-cant-away-wife-brummie18007999 (accessed on 4 March 2021).

Bateman, S. 2020. Coronavirus gives sex cam girls a business boom as randy customers stuck at home. *Daily Star*, March 15. Available online: https://www.dailystar.co.uk/news /worldnews/coronavirus-gives-sex-cam-girls-21696804 (accessed on 4 September 2020).

Bateman, S. 2020. Older sex workers taught cam sex by younger women during coronavirus lockdown. *Daily Star*, May 6. Available online: https://www.dailystar.co.uk/news/w orld-news/older-sexworkers-taught-cam-21984276 (accessed on 4 September 2020).

Boseley, M. 2020. 'Everyone and their mum is on it': OnlyFans booms in popularity during the pandemic. *The Guardian*, December 22. Available online: https://www.theguard ian.com/technology/2020/dec/23/everyone-and-their-mum-is-on-itOnlyFans-boomed-in -popularity-during-the-pandemic (accessed on 2 January 2021).

Boseley, M. 2020. 'The sex industry is not pandemic-proof': workers in Australia faced with impossible choices. *The Guardian*, November 7. Available online: https://www.thegua rdian.com/society/2020/nov/08/the-sex-industry-is-not-pandemic-proof-workers-in-au stralia-faced-with-impossible-choices (accessed on 2 January 2021).

Brett, A. 2020. OnlyFans: why are A-list celebrities joining a pornographic site? *Telegraph*, August 29. Available online: https://www.telegraph.co.uk/on-demand/0/Only Fans-a-list-celebrities-joiningpornographic-site/ (accessed on 4 September 2020).

Brown, A. 2020. Scots women warned sex webcamming is not easy way to make money during lockdown. *Daily Record*, May 8. Available online: https://www.dailyrecord.

co.uk/news/scottishnews/women-turning-sex-webcamming-make-21993210 (accessed on 4 September 2020).

Brown, A., and Moran, M. 2020. Warning issued as increase in women turning to webcam sex to make cash during lockdown. *Daily Star*, May 8. Available online: https://www.dailystar.co.uk/news/latest-news/warning-issued-increase-women-turning-21994640 (accessed on 4 March 2021).

Cavanagh, M. 2021. 'HAVE TO SURVIVE' Desperate students selling sex & signing up to OnlyFans as traditional bar jobs dry up and Covid forced shops to close. *The Scottish Sun*, April 11. Available online: https://www.thescottishsun.co.uk/news/6951748/desperate-students-selling-sex-onlyfans-covid/ (accessed on 4 March 2021).

Das, S. 2020. Meet the king of homemade porn—a banker's son making millions. *The Times*, July 26. Available online: https://www.thetimes.co.uk/article/meet-the-king-of-homemade-porn-a-bankersson-making-millions-z9vhq9c9s (accessed on 4 March 2021).

Davies, A. 2020. OnlyFans: Digital footprint concern raised over images. *BBC*, December 9. Available online: https://www.bbc.co.uk/news/uk-wales-55130695 (accessed on 4 March 2021).

Davis, K. 2021. Nurse fed up with being 'taken for granted' during Covid crisis earns FIVE times her salary as an erotic cam girl. *The Scottish Sun*, February 11. Available online: https://www.thescottishsun.co.uk/news/6667242/nurse-covid-earns-five-times-salary-erotic-cam-girl/ (accessed on 4 March 2021).

Drolet, G. 2020. The sex workers working from home. *The Independent*, April 19. Available online: https://www.independent.co.uk/life-style/coronavirus-sex-work-porn-home-webcam-stream-onlyfans-a9464326.html (accessed on 4 March 2021).

Dunstan, G. 2020. Coronavirus: How do sex workers keep safe during pandemic?. *BBC*, May 19. Available online: https://www.bbc.co.uk/news/uk-wales-52706165 (accessed on 4 March 2021).

Foster, S. 2020. Cam girls say they can earn more than £2,000 a week–here's how it works. *Daily Star*, April 22. Available online: https://www.dailystar.co.uk/love-sex/cam-girls-say-can-earn-21907657 (accessed on 4 March 2021).

Foster, S. 2020. Cam girls fear they'll 'turn into sex robots' as demand soars during lockdown. *Daily Star*, April 21. Available online: https://www-proquest-com.ezproxy.lib.bbk.ac.uk/europeannews/docview/2392830397/5B7370F9BB454281PQ/25?accountid=8629 (accessed on 4 March 2021).

Husselbee, R. 2021. LOVE BUGS Horny men with secret covid fetish pay me £4k a month to cough and sneeze on FaceTime, reveals page 3 girl. *The Sun*, May 6. Available online: https://www.thesun.co.uk/news/14868194/4k-coughing-sneezing-covid-fetish-facetime/ (accessed on 20 May 2021).

Jones, L. 2020. OnlyFans: 'I started selling sexy photos online after losing my job'. *BBC*, July 15. Available online: https://www.bbc.co.uk/news/business-53338019 (accessed on 4 March 2021).

Kennedy, P. 2020. Heartbreaking stories of 'desperate' women who have turned to selling sex online to survive during pandemic. *Sunderland Echo*, December 4. Available online: https://www-proquest-com.ezproxy.lib.bbk.ac.uk/europeannews/docview/2467372475/fulltext/5DEED86231E84FC5PQ/1?accountid=8629 (accessed on 4 March 2021).

Kirby, D. 2020. Rise in online sex work 'linked to Covid poverty': SOCIETY. *i*. December 4. Available online: https://www-proquest-com.ezproxy.lib.bbk.ac.uk/europeannews/docview/2466701635/358502AE2C6147D8PQ/26?accountid=8629m (accessed on 4 March 2021).

Knox, M. 2020. 'I swapped my office job for online sex work in lockdown . . . but my fiance loves it'. *The Scottish Sun*, December 6. Available online: https://www.thescottishsun.co.uk/news/6371062/sexwork-online-office-job-fiance/ (accessed on 4 March 2021).

Manavis, S. 2020. How the rich and famous stole OnlyFans from sex workers. *New Statesman*, September 18. Available online: https://www.newstatesman.com/science-tech/social-media/2020/09/rich-famous-onlyfans-changing-sex-workers-left-behind-bella-thorne-caroline-calloway-beyonce (accessed on 4 March 2021).

Meneely, G. 2020. HOOKERS MOVE SEX BIZ ONLINE: ESCORTS USE WEB TO SERVICE CLIENTS Irish escorts start working from home. *The Sun*, April 19. Available online: https://www-proquestcom.ezproxy.lib.bbk.ac.uk/europeannews/docview/239 1795217/358502AE2C6147D8PQ/86?accountid=8629 (accessed on 4 March 2021).

Lewis, F. 2020. Young Welsh scientist says posting explicit content online during pandemic has helped to pay off debts. *Wales Online*, December 10. Available online: https://www.walesonline.co.uk/news/wales-news/young-welsh-scientist-says-posting19424426 (accessed on 4 March 2021).

Martins, S. 2020. What it's like to be a sex worker in lockdown. *Metro*, May 22. Available online: https://metro.co.uk/2020/05/22/what-like-sex-worker-lockdown-126 92175/ (accessed on 4 March 2021).

Perry, L. 2020. How OnlyFans became the porn industry's great lockdown winner–and at what cost. *New Statesman*, November 4. Available online: https://www.newstatesman.com/sciencetech/social-media/2020/11/how-OnlyFans-became-porn-industry-s-great-lockdown-winnerand-what (accessed on 4 March 2021).

Pochin, C. 2020. 'I lost my job during lockdown so became a cam girl to pay the bills'. *Mirror*, May 5. Available online: https://www.mirror.co.uk/lifestyle/sex-relationships/sex/i-lost-job-during-lockdown21977968 (accessed on 4 March 2021).

Price, M.L. 2021. Pandemic makes prostitution taboo in Nevada's legal brothels. *Independent*, February 20. Available online: https://www.independent.co.uk/news/pandemic-makes-prostitution-taboo-innevadas-legal-brothels-pandemic-sex-workers-nevada-brothels-prostitutes-b1805097.html (accessed on 4 March 2021).

Robinson, A. 2020. Coronavirus: More students are turning to sex work during COVID-19 pandemic. *Sky News*, September 10. Available online: https://news.sky.com/story/coronavirus-more-students-are-turning-to-sex-work-during-COVID-19-pandemic-12066700 (accessed on 4 March 2021).

Sharpe, A. 2021. Desperate students turn to sex and OnlyFans after lockdown kills pub and shop jobs. *Mirror*, April 10. Available online: https://www.mirror.co.uk/news/uk-news/desperate-students-sell-sex-pay-23890764 (accessed on 4 March 2021).

Shehadi, S., & Partington, M. 2020. Coronavirus: Offline sex workers forced to start again online. *BBC*, April 7. Available online: https://www.bbc.co.uk/news/technology-52 183773 (accessed on 4 March 2021).

Sims, P., & Perrie, R. 2020. Sex workers demand aid from government because married clients 'can't sneak off' during coronavirus lockdown. *The Sun*, April 14. Available online: https://www.thesun.co.uk/news/11391338/sex-workers-aid-government-married-clientssneak-coronavirus/ (accessed on 4 March 2021).

Wade-Palmer, C. 2020. Porn stars on coronavirus lockdown to swap sex scenes for solo webcam streams. *Daily Star*, March 20. Available online: https://www.dailystar.co.uk/news/latest-news/porn-stars-coronavirus-lockdown-swap-21722358 (accessed on 4 March 2021).

Withers, L., & King, L. 2021. Mum who lost job in lockdown set to become OnlyFans millionaire in just one year. *Mirror*, March 30. Available online: https://www.mirror.co.uk/news/uk-news/mum-who-lost-job-lockdown-23819036 (accessed on 4 March 2021).

Younan, C. 2020. Cam girl says 'men are gagging for it' more as coronavirus boosts business. *Daily Star*, July 19. Available online: https://www-proquest-com.ezproxy.lib.bbk.ac.uk/europeannews/docview/2424819336/5B7370F9BB454281PQ/5?accountid=8629 (accessed on 4 March 2021).

Younan, C. 2020. Cam girl raves about new sex trend that makes 'imaginations run wild'. *Daily Star*, August 19. Available online: https://www.dailystar.co.uk/love-sex/cam-girl-says-new-sex-22512086 (accessed on 4 March 2021).

## Note

[1] All of the included excerpts were extracted in vivo from the analysed articles.

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
