# Peer review of "‘Cam Girls and Adult Performers Are Enjoying a Boom in Business’: The Reportage on the Pandemic Impact on Virtual Sex Work"

_socsci, doi:10.3390/socsci12020062_

Round 1

Reviewer 1 Report

This is a fascinating topic, which definitely merits further study.  however grounded theory is a weak analytic lens which contributes to the fact that the findings are underdeveloped, (what is 'empowerment' ) along with the fact that authors do not engage with much research on mediatized sex work and agency - ie Kira Hall, Fareen Pervez, Mi-Cha Flubacher.

recommend authors conduct a more systematic and critical analysis, cite more literature to offer an academic contribution instead of the truism that sx workers - like any workers - should not be exploited.

Author Response

Reviewer 1

We thank Reviewer 1 for their comments on this fascinating topic meriting further study. 

  1. More evidence is provided to justify the choice of GT as an appropriate form of analysis for this exploratory study and also highlighted why is an appropriate form of analysis in the methods section too.
  2. We have developed the findings further - at quite some length, by including another study initially intended for individual publication, and as a result the study has now doubled in size. This also adds inter-rater reliability to the findings as the analysis was carried out by two researchers, and the findings of the two studies combine to show the robustness of methods and of the findings. Because news narratives and counter-narratives are forged in news media they offer a valuable source for analysis. They also offer an immediacy that many 'hindsight' research interviews cannot. Researchers of sex work understandably make attributions about what is written in the press about sex work so it is important to analyse the press coverage itself, not least to prevent fundamental misattributions about what is said or written about in the news media. This is a critical strength of this study. The research did not seek to explore whether the coverage was positive or negative per se but rather by utitilizing the immediacy of the content (made in real time) to harness the content itself, it shows in a systematic way, what that reportage contained. These findings will be an important comparator to researcher interviews to illustrate their differences.
  3. Authors now engage with much research on mediatized sex work and agency - ie Kira Hall, Fareen Parvez, and Mi-Cha Flubacher. As regards the question of empowerment, we now address this using Halls' (1995) notion of empowerment within patriarchial privilege of economic power and duly contextualize the study more completely.
  4. The authors conducted a more systematic and critical analysis, have cited much more relevant literature to offer an academic contribution on the nuances of the mediatization of online sex work during the pandemic.

Reviewer 2 Report

The paper entitled: “Cam Girls and Adult Performers are Enjoying a Boom in Business’: A Grounded Theory Media Study of the Pandemic Impact on Virtual Sex Work” aims at understanding the impact of Covid-19 on the digital sex industry by exploring the ways in which the coronavirus pandemic has affected the lifestyle of digital sex workers. At the methodological level, this study, conducted between January 2021 and May 2021, is based on the content analysis of 20 newspaper articles released in the United Kingdom during the pandemic. The authors counterbalanced the right-wing, left-wing and centrist political alignment of the newspapers. On the other hand, the paper’s content, except 2, was derived from interviews with digital sex workers.

            This is a very interesting paper that could contribute to increasing public awareness of the digital sex industry and to the destigmatisation and mainstreaming of Virtual Sex Work, apparently a source of income for a growing number of women in the UK.

However, I have my doubts about the representativeness of the experiences of digital sex workers reported in this paper. The sample is very small, and the authors do not study primary data, but secondary data. This is a secondhand study of interviews analyzed and published by other authors with different political alignments. Therefore, I strongly advise the authors to continue this line of research by engaging in in-depth interviews with digital sex workers.

On the one hand, I think that this paper does not need ethical approval from an ethics committee because the authors are studying secondary data, they have not interviewed sex workers.

On the other hand, the authors could improve the analysis of the data studied by examining the different approaches to Virtual Sex Work in right-wing, left-wing, and centrist UK newspapers. There are no differences in the approaches to Virtual Sex Work among right-wing, left-wing, and centrist UK newspapers?

Author Response

We thank Reviewer 2 for their insights and we have addressed their comments in the following ways.

  1. The data's representativeness of what was reported is greatly strengthened by the enlargement of the study and the addition of another study (Study 2 which was initially intended for publication elsewhere) as well as the inter-rater reliability of using two sets of analysis carried out by two researchers independently of each other. Its main strength, however, is in the theoretical model that emerges and patterns that emerge in the reportage. Further strengths are also in the exploration of the mediatization of online sex work. The paper now clarifies that it does not aim for generalizability nor does it make any claim for representativeness of the findings as to what sex workers did or did not do, but rather the development of a model that illustrates how the media reports the issue. It is therefore, the notion of 'transferability' (Henwood & Pidgeon, 1995; Willig, 2005) that is the aim and we make apparent that because

Because news narratives and counter-narratives are forged in news media they offer a valuable source for analysis. They also offer an immediacy that many 'hindsight' research interviews cannot. Researchers of sex work understandably make attributions about what is written in the press about sex work so it is important to analyse the press coverage itself, not least to prevent fundamental misattributions about what is said or written about in the news media. This is a critical strength of this study. The research did not seek to explore whether the coverage was positive or negative per se but rather by utitilizing the immediacy of the content (made in real time) to harness the content itself, it shows in a systematic way, what that reportage contained. These findings will be an important comparator to researcher interviews to illustrate their differences.

  1. The sample was expanded from 20 to 40 reports to achieve saturation of the reportage over that period of time.

  1. The authors may continue the line of research by engaging in-depth interviews with digital sex workers, and these findings would contribute to the development of those interview themes. It may also be used to explore the online platforms as economic units to explore the 'increased workload vs. increased competition' nuances that emerged in the findings and examine the prevalence of these and other novel findings that contribute to the literature. They do, however, now stand alone for the reasons above.

  1. The paper mentions ethical approval because the protocol at the institution where it was carried out is such that all studies, including those using online or secondary data, must have approval from the ethics committee. The ethical strength in not taking further time away from sex workers at a critical time for their incomes is mentioned. The ethics section has, however, been reduced